# Self-supervise, Refine, Repeat:
# Improving Unsupervised Anomaly Detection

**Jinsung Yoon, Kihyuk Sohn, Chun-Liang Li, Sercan Ö. Arik, Chen-Yu Lee, Tomas Pfister**
*{jinsungyoon, kihyuks, chunliang, soarik, chenyulee, tpfister}@google.com*
*Google Cloud AI*

**Reviewed on OpenReview:** *https://openreview.net/forum?id=b3v1UrtF6G*

## Abstract

Anomaly detection (AD), separating anomalies from normal data, has many applications across domains, from security to healthcare. While most previous works were shown to be effective for cases with fully or partially labeled data, that setting is in practice less common due to labeling being particularly tedious for this task. In this paper, we focus on fully *unsupervised* AD, in which the entire training dataset, containing both normal and anomalous samples, is unlabeled. To tackle this problem effectively, we propose to improve the robustness of one-class classification trained on self-supervised representations using a data refinement process. Our proposed data refinement approach is based on an ensemble of one-class classifiers (OCCs), each of which is trained on a disjoint subset of training data. Representations learned by self-supervised learning on the refined data are iteratively updated as the data refinement improves. We demonstrate our method on various unsupervised AD tasks with image and tabular data. With a 10% anomaly ratio on CIFAR-10 image data / 2.5% anomaly ratio on Thyroid tabular data, the proposed method outperforms the state-of-the-art one-class classifier by 6.3 AUC and 12.5 average precision / 22.9 F1-score.

## 1 Introduction

Anomaly detection (AD), the task of distinguishing anomalies from normal data, plays crucial role in many real-world applications such as detecting faulty products from vision sensors in manufacturing, fraudulent behaviors at credit card transactions, or adversarial outcomes at intensive care units such as death, heart attack, or blood poisoning.

AD has been considered under various settings based on the availability of negative (normal) and positive (anomalous) data and their labels at training, as overviewed in Sec. 2. Each application scenario is dominated by different challenges. When all positive and negative samples are available along with their labels (Fig. 1a), the problem can be treated as supervised classification and the dominant challenge becomes the imbalance in label distributions (Chawla et al., 2002; Estabrooks et al., 2004; Hwang et al., 2011; Barua et al., 2012; Lee, 2000; Liu et al., 2007). When only negative labeled data are available (Fig. 1b), the problem is 'one-class classification' (Schölkopf et al., 1999; Tax & Duin, 2004; Ruff et al., 2018; Hendrycks et al., 2018; Golan & El-Yaniv, 2018; Sohn et al., 2021; Li et al., 2021). Various works have also extended approaches designed for these to settings with additional unlabeled data (Fig. 1c,d,e) (Zhang & Zuo, 2008; Blanchard et al., 2010; Görnitz et al., 2013; Ruff et al., 2020) in a semi-supervised setting. While there exist many prior works in these settings, most of them rely on some labeled data, which is not desirable in many application scenarios.

Unsupervised AD, on the other hand, poses unique challenges in the absence of any labeled data information, and a straightforward adaption of methods developed with the assumption of labeled data would be suboptimal. For example, some recent studies (Zong et al., 2018; Bergman & Hoshen, 2019) have applied one-class classifiers (OCCs) that are known to yield the state-of-the-art performance when trained on negative samples (Golan & El-Yaniv, 2018; Hendrycks et al., 2018; Bergman & Hoshen, 2019; Sohn et al., 2021; Li et al., 2021) to

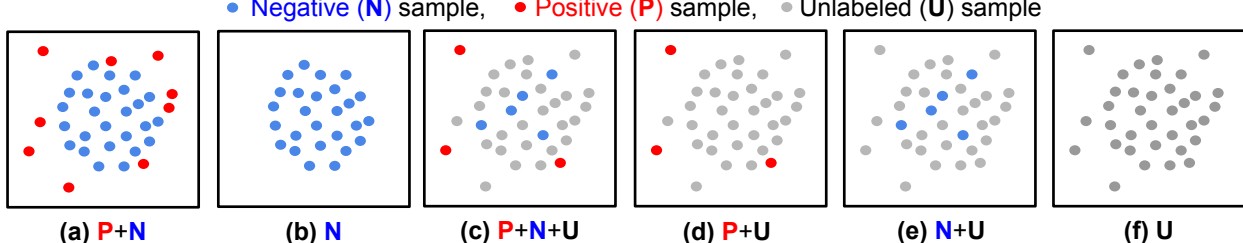

Figure 1: Anomaly Detection (AD) problem settings. Blue and red dots are for **labeled** negative (normal) and positive (anomalous) samples, respectively. Grey dots denote **unlabeled** samples. While previous works mostly focus on supervised (a, b) or semi-supervised (c, d, e) settings, we tackle an AD problem using only unlabeled data (f) that may contain both negative and positive samples.

unsupervised AD, but their performance for unsupervised AD (when the unlabeled data contain both positive and negative samples) has been significantly degraded. Fig. 2 illustrates this, showing the unsupervised AD performance of state-of-the-art Deep OCCs (Sohn et al., 2021) with different anomaly ratios in unlabeled training data – the average precision significantly drops even with a small contamination ratio (2%) of the training data.

Our framework SRR (**S**elf-supervise, **R**efine, **R**epeat), overviewed in Fig. 3, brings a data-centric[1] approach to unsupervised AD with the principles of self-supervised learning without labels and iterative data refinement based on the agreement of OCC outputs. We propose to improve the state-of-the-art performance of OCCs, e.g. (Sohn et al., 2021; Li et al., 2021), by refining the unlabeled training data so as to address the challenges elaborated above. SRR iteratively trains deep representations using refined data while improving the refinement of unlabeled data by excluding potentially-positive (anomalous) samples. For the data refinement process, we employ an ensemble of OCCs, each of which is trained on a disjoint subset of unlabeled training data. The samples are declared as normal if there is a consensus between all the OCCs. The refined training data are used to train the final OCC to generate the anomaly scores in the unsupervised setting.

Most prior AD works (Golan & El-Yaniv, 2018; Hendrycks et al., 2018; Sohn et al., 2021; Li et al., 2021) assume that the data contain entirely negative samples, which makes them not truly unsupervised as they require having humans to do the data

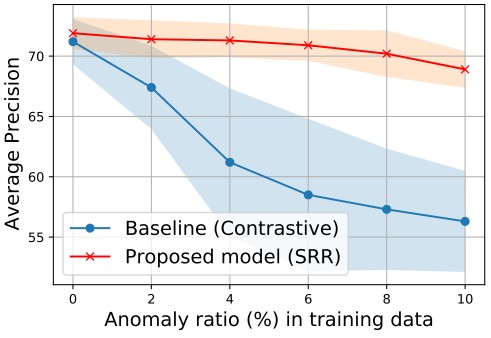

Figure 2: Performance of the proposed model (SRR) and a baseline OCC using contrastive representation (Sohn et al., 2021) on CIFAR-10 with different anomaly ratios in the training data.

filtering. Most prior works are not actually designed for fully-unsupervised anomaly detection. Some prior unsupervised AD works (Ruff et al., 2018; Zong et al., 2018; Bergman & Hoshen, 2019) consider evaluating on an unsupervised setting where there exist a small percentage of anomalous samples in the training data, i.e. operating in 'truly' unsupervised setting without having the need for humans to do any filtering in the training data. However, these methods often suffer from significant performance degradation as the ratio of anomalous sample ratio increases (see Sec. 4.2). Our method distinguishes from the literature by bringing a data-centric approach to unsupervised AD beyond the model-centric approaches. We show the value of SRR in improving robustness of the performance as the anomaly ratio increases, as shown in Fig. 2.

We conduct extensive experiments across various datasets from different domains, including semantic AD (CIFAR-10 (Krizhevsky & Hinton, 2009), Dog-vs-Cat (Elson et al., 2007)), real-world manufacturing visual AD use case (MVTec (Bergmann et al., 2019)), and real-world tabular AD benchmarks (e.g., detecting

---

[1]Definitions of the data-centric approaches can be found here (https://spectrum.ieee.org/andrew-ng-data-centric-ai).

medical or network anomalies). We consider methods with both shallow (Schölkopf et al., 1999; Liu et al., 2008) and deep (Sohn et al., 2021; Li et al., 2021; Bergman & Hoshen, 2019) models. We evaluate models at different anomaly ratios of unlabeled training data and show that SRR significantly boosts performance. For example, in Fig. 2, SRR improves more than 15.0 average precision (AP) with a 10% anomaly ratio compared to a state-of-the-art one-class deep model (Sohn et al., 2021) on CIFAR-10. Similarly, on MVTec, SRR retains a strong performance, dropping less than 1.0 AUC with 10% anomaly ratio, while the best existing OCC (Li et al., 2021) drops more than 6.0 AUC. The contributions of this paper are summarized below.

- We propose a novel data-centric framework, SRR, for unsupervised anomaly detection using the ensemble of one-class classifiers as a data refinement module.
- The proposed framework is a model-agnostic approach that can be applicable on top of any anomaly detection framework, considering different ways of applying self-supervised representation learning or one-class classification.
- SRR achieves significant robustness improvements with various anomaly ratios – in other words, the users do not need to worry about manually filtering the possible anomalies from the training data to minimize contamination. We demonstrate the superior performances on multiple tabular and image datasets.

## 2 Related Work

There are various existing works under different settings described in Fig. 1:

**Learning from both positives & negatives setting** is often considered as a supervised binary classification problem. The challenge arises due to the imbalance in label distributions as positive samples are rare. As summarized in (Branco et al., 2015), to address this, over-/under-sampling (Chawla et al., 2002; Estabrooks et al., 2004), weighted optimization (Hwang et al., 2011; Barua et al., 2012), synthesizing data of minority classes (Lee, 2000; Liu et al., 2007), and hybrid methods (Galar et al., 2011) have been studied.

**Learning only from the negatives setting** is often converted to a one-class classification (OCC) problem, with the goal of finding a decision boundary that includes as many one-class samples as possible. Shallow models for this setting include one-class support vector machines (Schölkopf et al., 1999) (OC-SVM), support vector data description (Tax & Duin, 2004) (SVDD), kernel density estimation (KDE) (Latecki et al., 2007), and Gaussian density estimation (GDE) (Reynolds, 2009). There are also auto-encoder based models (Zhou & Paffenroth, 2017) that treat the reconstruction error as the anomaly score. Deep learning based OCCs have been developed, such as Deep OCC (Ruff et al., 2018), geometric transformation (Golan & El-Yaniv, 2018), or outlier exposure (Hendrycks et al., 2018). Noting the degeneracy or inconsistency of learning objectives of existing end-to-end trainable Deep OCCs, (Sohn et al., 2021) propose a deep representation OCC, a two-stage framework that learns self-supervised representations (Komodakis & Gidaris, 2018; Chen et al., 2020) followed by shallow OCCs. That work is extended for texture anomaly localization with CutPaste (Li et al., 2021). Robustness against very low anomaly ratios of these in unsupervised setting is explored in (Zong et al., 2018; Bergman & Hoshen, 2019).

**The semi-supervised learning setting** utilizes a small set of labeled samples and large set of unlabeled samples to distinguish anomalies from normal data. Depending on which labeled samples are given, this setting can be split into three sub-categories. When only some positive/negative labeled samples are provided, we denote that as a PU (positive + unlabeled) / NU (negative + unlabeled) setting. Most previous works in semi-supervised AD settings focus on the NU setting where only some of the normal labeled samples are given (Muñoz-Marí et al., 2010; Song et al., 2017; Akcay et al., 2018). The PNU (positive + negative + unlabeled) setting is the more general semi-supervised setting where subsets of both positive and negative labeled samples are given. Deep SAD (Ruff et al., 2020) and SU-IDS (Min et al., 2018) belong to this category. We show significant outperformance of SRR compared to Deep SAD, on multiple benchmark datasets without using any labeled data (see Sec. 4.2).

**The unlabeled setting** has received relatively less attention despite its significance in minimizing labeling costs. The popular methods for this setting include isolation forest (Liu et al., 2008) and local outlier factor (Breunig et al., 2000). These are difficult to scale, and less compatible with recent advances in representation learning. While OCCs, such as OC-SVM, SVDD, or their deep counterparts, can also be

applied to unlabeled settings by assuming the data are all negative, and the robustness of those methods has been demonstrated in part (Zong et al., 2018; Bergman & Hoshen, 2019). In practice, we observe a significant performance drop with a high anomaly ratio, shown in Fig. 2. In contrast, our proposed framework is able to maintain high performance across different anomaly ratios.

**Data refinement** has been applied to AD in some prior work. (Pang et al., 2020; Beggel et al., 2019) generate pseudo-labels using binary classification and OC-SVM for data refinement to boost the consequent AD performance in unsupervised settings. (Mohseni et al., 2021; Xia et al., 2015; Zhou & Paffenroth, 2017; Lai et al., 2019; Berg et al., 2019) use the reconstruction errors of the auto-encoder as an indicator for removing possible anomalies. (Feng et al., 2021) and (Meng et al., 2021) use data refinement for AD in supervised and semi-supervised settings. SRR differentiates from (Xia et al., 2015; Beggel et al., 2019; Pang et al., 2020) in multiple key ways:

- (Xia et al., 2015; Beggel et al., 2019) are based on auto-encoders and they directly utilize the reconstruction errors as informative signals for AD and iterative data refinement. However, prior work (Ren et al., 2019) discovered that the reconstruction is not a good informative signal for outlier detection. Also as shown in Fig. 5, the performance of DAE (reconstruction errors based AD) is much worse than alternatives.
- SRR utilizes ensemble learning for data refinement to improve the robustness which is critical in unsupervised AD. On the other hand, (Xia et al., 2015; Beggel et al., 2019; Pang et al., 2020) utilize a single model for data refinement. As can be seen in Fig. 8a, the impact of the ensemble model for data refinement is significant.
- (Xia et al., 2015; Beggel et al., 2019; Pang et al., 2020) directly utilize the (pseudo) abnormal samples for model training in addition to (pseudo) normal samples. For instance, the model in (Pang et al., 2020) is trained via two-class ordinal regression where two classes come from (pseudo) normal and (pseudo) abnormal samples. Directly relying on the (imperfectly) labeled abnormal samples can be harmful for AD due to the overfitting problems and high False Positive Rates (FPR) in (pseudo) abnormal samples.[2]
- (Xia et al., 2015; Pang et al., 2020) are based on the end-to-end AD models which are empirically less accurate than two-stage models (Sohn et al., 2021).

**Self-training** (Scudder, 1965; McLachlan, 1975) is an iterative training mechanism using predicted pseudo labels as targets. It has regained popularity recently with its successful results in semi-supervised image classification (Berthelot et al., 2019; Sohn et al., 2020; Xie et al., 2020). To improve the quality of pseudo labels, employment of an ensemble of classifiers has also been studied. (Brodley et al., 1996) trains an ensemble of classifiers with different classification methods to make a consensus for noisy label verification, while co-training (Blum & Mitchell, 1998) trains multiple classifiers, each of which is trained on the distinct views, to supervise other classifiers. Co-teaching (Han et al., 2018) and DivideMix (Li et al., 2019) share similar ideas – they both train multiple deep neural networks on separate data batches to learn different decision boundaries, thus, becoming useful for noisy label verification. While sharing a similarity, SRR has clear differences from the previous works – SRR performs *iterative training* with data refinement (with robust *ensemble* methods) and self-supervised learning for *unsupervised* AD.

## 3 Proposed Framework

**S**elf-supervise, **R**efine, and **R**epeat (SRR) is an iterative training framework, where we refine the data (Sec. 3.1) and update the representation with the refined data (Sec. 3.2), followed by OCC training on refined representations. Fig. 3 overviews the framework and Algorithm 1 provides the pseudo-code.

**Notation.** We denote the training data as $\mathcal{D} = \{\mathbf{x}_i\}_{i=1}^N$ where $\mathbf{x}_i \in \mathcal{X}$ and $N$ is the number of training samples. $y_i \in \{0, 1\}$ is the corresponding label to $\mathbf{x}_i$, where 0 denotes normal (negative) and 1 denotes anomaly (positive). Note that labels are not provided in the unsupervised setting.

---

[2]For instance, with 80% of the recall (for the abnormal sample discovery), the precision (for the abnormal sample discovery) is 28.2% for CIFAR-10 and 31.7% with MVTec (with 6% abnormal ratio) using SRR framework – the majority of the (pseudo) abnormal samples are actually the normal samples (71.8% for CIFAR-10 and 68.3% for MVTec datasets). This would be a strong evidence that directly utilizing the (pseudo) abnormal samples for training can be harmful.

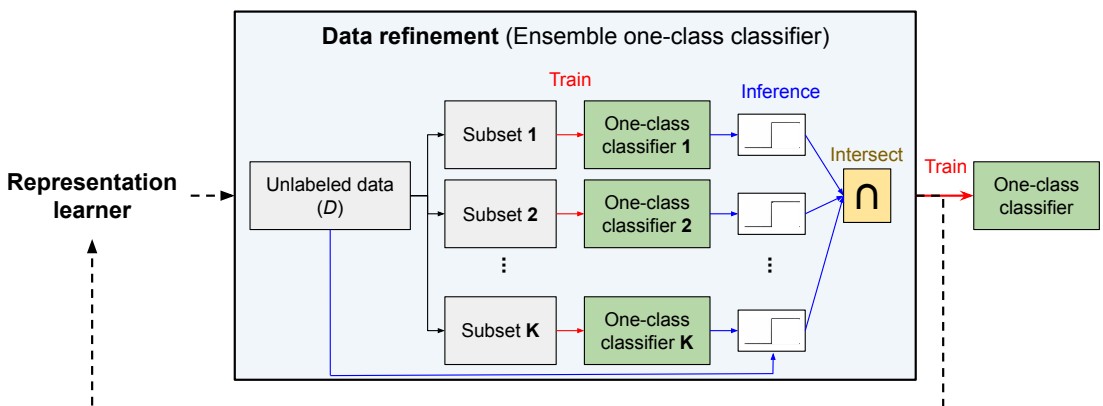

Figure 3: Block diagram of SRR composed of representation learner (Sec. 3.2), data refinement (Sec. 3.1), and final OCC blocks. The representation learner updates the corresponding deep neural network models using refined data from the data refinement block. Data refinement is implemented by an ensemble of OCCs, each of which is trained on $K$ disjoint subsets of unlabeled training data. Samples predicted as normal by all classifiers are retained in the refined data, and are used to update the representation learner and final OCC. The training process of SRR is repeated until the convergence of the representation learner's loss (if no improvement is observed for 5 epochs). Convergence graphs are provided in the Appendix A.3.

Let us denote a feature extractor as $g : \mathcal{X} \to \mathcal{Z}$. $g$ may include any data preprocessing functions, an identity function (if raw data is directly used for one-class classification), and learned or learnable representation extractors such as deep neural networks. Let us define an OCC as $f : \mathcal{Z} \to [-\infty, \infty]$ that outputs anomaly scores given the input features $g(\mathbf{x})$, such that the higher the score $f(g(\mathbf{x}))$ is, the more anomalous the sample $\mathbf{x}$ would be predicted as. The binary AD is made after thresholding: $\mathbb{1}\big(f(g(\mathbf{x})) \geq \eta\big)$.

## 3.1 Data refinement

A naive way to generate pseudo labels for unlabeled data would be to construct an OCC on raw data or learned representations as in (Sohn et al., 2021), pick a threshold for the anomaly score and obtain a binary label for samples as normal vs. anomalous. As we update the model with refined data that exclude samples that are predicted to be anomalous, it is important to generate pseudo labels for training data accurately. To this end, instead of training a single classifier (unlike most previous works on data refinement for AD (Pang et al., 2020; Beggel et al., 2019; Zhou & Paffenroth, 2017)), we train an ensemble of $K$ OCCs and aggregate their predictions to generate pseudo labels.

We illustrate the data refinement block in Fig. 3 and as REFINEDATA in Algorithm 1. Specifically, we randomly divide the unlabeled training data $\mathcal{D}$ into $K$ disjoint subsets $\mathcal{D}_1, ..., \mathcal{D}_K$, and train $K$ different OCCs $(f_1, ..., f_K)$ on corresponding subsets $(\mathcal{D}_1, ..., \mathcal{D}_K)$. Then, we estimate a binary pseudo-label of the data $\mathbf{x}_i \in \mathcal{D}$ as follows:

$$\hat{y}_i = 1 - \prod_{k=1}^{K} \left[ 1 - \mathbb{1}\big(f_k(g(\mathbf{x}_i)) \geq \eta_k\big) \right] \tag{1}$$

$$\eta_k = \max \eta \ \text{ s.t. } \ \frac{100}{N} \sum_{i=1}^{N} \mathbb{1}\big(f_k(g(\mathbf{x}_i)) \geq \eta\big) \geq \gamma \tag{2}$$

where $\mathbb{1}(\cdot)$ is the indicator function that outputs 1/0 if the input is True/False. $f_k(g(\mathbf{x}_i))$ represents an anomaly score of $\mathbf{x}_i$ for an OCC $f_k$. $\eta_k$ in Eq. 2 is a threshold determined as a $\gamma$ percentile of the anomaly score distribution $\{f_k(g(\mathbf{x}_i))\}_{i=1}^{N}$.

To interpret Eq. 1, $\mathbf{x}_i$ is predicted as normal, i.e. $\hat{y}_i = 0$, if all $K$ OCCs predict it as normal. While this may be too strict and potentially reject many true normal samples in the training set, we find that empirically, it is

---

**Algorithm 1** SRR: Self-supervise, Refine, Repeat.

---

**Input**: Train data $\mathcal{D} = \{\mathbf{x}_i\}_{i=1}^N$, Ensemble count $(K)$, threshold $(\gamma)$
**Output**: Refined data $(\hat{\mathcal{D}})$, trained OCC $(f)$, feature extractor $(g)$

1: **function** REFINEDATA$(\mathcal{D}, g, K, \gamma)$
2:      Train OCC models $\{f_k\}_{k=1}^K$ on $\{\mathcal{D}_k\}_{k=1}^K$, $K$ disjoint subsets of the training data $\mathcal{D}$.
3:      Compute thresholds $\eta_k$'s for $\gamma$ percentile of anomaly distributions using Eq. 2.
4:      Predict binary labels $\hat{y}_i$ using Eq. 1.
5:      Return $\hat{\mathcal{D}} = \{\mathbf{x}_i : \hat{y}_i = 0, \mathbf{x}_i \in \mathcal{D}\}$.
6: **end function**
7: **function** SRR$(\mathcal{D}, K, \gamma)$
8:      Initialize the feature extractor $g$.
9:      **while** $g$ not converged **do**
10:          $\hat{\mathcal{D}} = $ REFINEDATA$(\mathcal{D}, g, K, \gamma)$.
11:          Update $g$ using $\hat{\mathcal{D}}$ with self-supervised objectives.
12:      **end while**
13:      $\hat{\mathcal{D}} = $ REFINEDATA$(\mathcal{D}, g, K, \gamma)$.
14:      Train an OCC model $(f)$ on refined data $(\hat{\mathcal{D}})$.
15: **end function**

---

critical to be able to exclude true anomalous samples from the training set. The effectiveness of employment of an ensemble of OCCs to improve the robustness against overfitting of OCCs is empirically shown in Sec. 5. More specifically, Fig 8a shows that the performance is much better for higher ensemble counts, compared to a single classifier training along with the feature extractor.

### 3.2 Representation update

SRR follows the idea of deep representation learning to perform OCCs on (Sohn et al., 2021), where in the first stage, a deep neural network is trained with self-supervised learning (such as rotation prediction (Golan & El-Yaniv, 2018), contrastive (Sohn et al., 2021), or CutPaste (Li et al., 2021)) to obtain meaningful representations of the data, and in the second stage OCCs are trained on these learned representations. Such a two-stage framework is shown to be beneficial as it prevents the 'hypersphere collapse' of the deep OCCs by the favorable inductive bias it brings with the architectural constraints (Ruff et al., 2018).

We propose to conduct self-supervised representation learning jointly with data refinement. More precisely, we train a feature extractor $g$ using $\hat{\mathcal{D}} = \{\mathbf{x}_i \mid \hat{y}_i = 0\}$, a subset of unlabeled data $\mathcal{D}$ that only includes samples whose predicted labels with an ensemble OCC from Sec. 3.1 are negative. We also update $\hat{\mathcal{D}}$ as we proceed with representation learning. Overall method is illustrated in Algorithm 1. Unlike previous works (Sohn et al., 2021; Li et al., 2021) that use the entire training data for representation learning, we empirically show that it is beneficial to refine the training data even for learning of the deep representations. Without representation refinement, the performance improvements of SRR are limited, as shown in Sec. 5. Lastly, to get predictions at test-time, we train an OCC on refined data $\hat{\mathcal{D}}$ on the updated representations by $g$ as in line 13-14 in Algorithm 1.

### 3.3 Unsupervised model selection

As SRR is designed for unsupervised AD, labeled validation data for hyperparameter tuning are typically not available and *the framework should enable robust model selection without any reliance on labeled data*. Here, we provide insights on how to select important hyperparameters, and later in Sec. 5.3, we perform sensitivity analyses for these hyperparameters.

Data refinement of SRR introduces two hyperparameters: the number of OCCs $(K)$ and the percentile threshold $(\gamma)$. There is a trade-off between the number of classifiers for the ensemble and the size of disjoint subsets for training each classifier. With large $K$, we aggregate prediction from many classifiers, each of which

may contain randomness from training. This comes at the cost of reduced performance per classifier as we use smaller subsets to train them. In practice, we find $K = 5$ works well across different datasets and anomaly ratios. $\gamma$ controls the purity and coverage of refined data – if $\gamma$ is large, and thus classifiers reject too many samples, the refined data could be more pure and contain mostly the normal samples; however, the coverage of the normal samples would be limited. On the other hand, with a small $\gamma$, the refined data may still contain many anomalies and the performance improvement with SRR would be limited. We empirically observe that SRR is robust to the selection of $\gamma$ when it is chosen from a reasonable range. In our experiments, we find 1-2 times of the true anomaly ratio to be often a reasonable choice. In other words, it is safer to use $\gamma$ higher than the expected true anomaly ratio. In some cases, the true anomaly ratio may not be available. For such scenarios, we propose Otsu's method (Sezgin & Sankur, 2004) to estimate the anomaly ratio of the training data for determining the threshold $\gamma$ (experiment results are in Sec. 6).

## 4 Anomaly Detection Performance

We evaluate the efficacy of our proposed framework for unsupervised AD tasks for tabular (Sec. 4.1) and image (Sec. 4.2) data types. We experiment varying ratios of anomaly samples in unlabeled training data and with different combinations of representation learning and OCCs. In Sec. 5, we provide performance analyses to better explain major constituents of the performance, as well as the sensitivity analyses.

**Implementation details:** To reduce the computational complexity of the data refinement block, we utilize a simple OCC such as GDE in the data refinement block. In a two-stage model, we only update the data refinement block at 1st, 2nd, 5th, 10th, 20th, 50th, 100th, 500th epochs. After 500 epochs, we update the data refinement block per each 500th epoch. Each experimental run is performed on a single V100 GPU. Additional discussions can be found in Appendix B.

### 4.1 Experiments on tabular data

**Datasets.** Following (Zong et al., 2018; Bergman & Hoshen, 2019), we test the performance of SRR on a variety of real-world tabular AD datasets, including network (KDDCup) and medical (Thyroid, Arrhythmia) AD from the UCI repository (Asuncion & Newman, 2007). We also use KDDCup-Rev, where the labels of KDDCup are reversed so that an attack represents the anomaly (Zong et al., 2018). To construct the data splits, we utilize 50% of normal samples for training. In addition, we hold out some anomaly samples (amounting to 10% of the normal samples) from the data. This allows to simulate unsupervised settings with an anomaly ratio of up to 10% of entire training set. Rest of the data is used for testing.[3] We conduct experiments using 5 random splits and 5 random seeds, and report the average and standard deviation of 25 F1-scores (with scale 0-100) for the performance metric.

**Models.** We focus on comparisons with GOAD (Bergman & Hoshen, 2019) (the state-of-the-art AD model in the tabular domain), with and without SRR. GOAD utilizes random transformation classification as the pretext task of self-supervised representation learning, and the normality score is determined by whether transformations are accurately included in the transformed space of the normal samples. We re-implement GOAD (Bergman & Hoshen, 2019) with a few modifications. First, instead of using embeddings to compute the loss, we use a parametric classifier, similarly to augmentation prediction (Sohn et al., 2021). Second, we follow the two-stage framework (Sohn et al., 2021) to construct deep OCCs. For the clean training data setting, our implementation achieves 98.0 for KDD, 95.0 for KDD-Rev, 75.1 for Thyroid, and 54.8 for Arrhythmia F1-scores, which are comparable to those reported in (Bergman & Hoshen, 2019). Please see Appendix C for formulation and implementation details.

We also include the comparisons with OC-SVM, with an radial basis function (rbf) kernel. For comparisons with other conventional baselines including Standard PCA (Wold et al., 1987), Robust PCA (Candès et al., 2011), and Local Outlier Factor (Breunig et al., 2000), please see the Appendix (we do not present them here as they are far worse than the state-of-the-art on the experimented datasets).

---

[3] Note that the experimental settings with contaminated training data in GOAD (Bergman & Hoshen, 2019) and DAGMM (Zong et al., 2018) are slightly different from ours. The contamination ratio in this paper is defined as the anomaly ratio over the entire training data, while their contamination ratio is defined as the anomaly ratio over all the anomalies in the dataset.

**Results.**

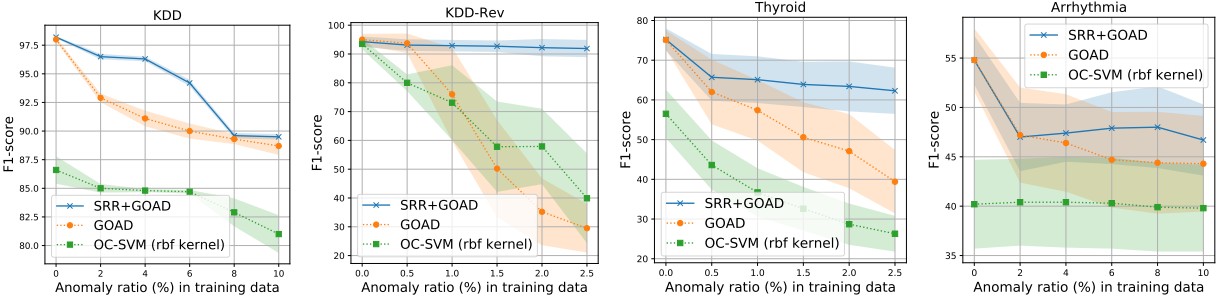

Figure 4: Unsupervised AD performance (F1-score) using OC-SVM (with rbf kernel), GOAD (Bergman & Hoshen, 2019), and GOAD with the proposed method (SRR) on four tabular datasets. Shaded areas represent the standard deviations.

Fig. 4 presents the results for SRR compared to the GOAD baseline, as well as OC-SVM. Overall, we observe superior AD performance with SRR consistently across datasets. For KDD-Rev, the performance of GOAD (without SRR) drops significantly even with a small anomaly ratio in training data, and beyond 1%, the drop becomes very significant. On the other hand, SRR can preserve the high anomaly detection performance despite the contamination in the training data, and at 2.5% anomaly ratio, we can observe almost the double F-score. The improvements on other datasets are smaller, but similarly we can observe significant F-score differences when the training data is contaminated with many anomaly samples. On two small-scale datasets, Thyroid & Arrhythmia, we observe more variance in the results due to data randomness when fitting the models, as expected. The improvements of SRR also highlights its data efficiency, as the proposed refinement mechanism can be effective even with a small subset of samples.

## 4.2 Experiments on image data

**Datasets.** We evaluate SRR on various visual AD benchmarks, including real-world manufacturing AD dataset (MVTec (Bergmann et al., 2019)) and semantic AD datasets (CIFAR-10 (Krizhevsky & Hinton, 2009), f-MNIST (Xiao et al., 2017), Dog-vs-Cat (Elson et al., 2007)). For CIFAR-10, f-MNIST, and Dog-vs-Cat datasets, samples from one class are set to be normal and the rest from other classes are set to be abnormal. Similar to the experiments on tabular data in Sec. 4.1, we swap a certain amount of the normal training data with anomalies given the target anomaly ratio. For MVTec, since there are no anomalous data available for training, we borrow 10% of the anomalies from the test set and swap them with normal samples in the training set. Note that 10% of samples borrowed from the test set are excluded from evaluation. For all datasets, we experiment with varying anomaly ratios from 0% to 10%.

We use AUC and average precision (AP) metrics to quantify the AD performance (with scale 0-100). When computing AP, we set the minority class of the test set as label 1 and majority as label 0. We run all experiments with 5 random seeds and report the average performance for each dataset across all classes.

**Models.** For semantic AD benchmarks, CIFAR-10, f-MNIST, and Dog-vs-Cat, we compare the SRR with two-stage OCCs (Sohn et al., 2021) using various representation learning methods, such as distribution-augmented contrastive learning (Sohn et al., 2021), rotation prediction (Komodakis & Gidaris, 2018; Golan & El-Yaniv, 2018) and its improved version, and denoising autoencoder. For MVTec benchmarks, we use CutPaste (Li et al., 2021) as the baseline and compare to its version with SRR integration. For both experiments, we use the ResNet-18 architecture, trained from random initialization, using the hyperparameters from (Sohn et al., 2021) and (Li et al., 2021). The same model and hyperparameter configurations are used for SRR with $K = 5$ classifiers in the ensemble. We set $\gamma$ as twice the anomaly ratio of training data. For 0% anomaly ratio, we set $\gamma$ as 0.5. Finally, a Gaussian Density Estimator (GDE) on learned representations is used as the OCC.

For comparisons with other conventional baselines including Standard PCA (Wold et al., 1987), Robust PCA (Candès et al., 2011), and Robust autoencoder (Zhou & Paffenroth, 2017), please see the Appendix (we do not present them here as they are far worse than the state-of-the-art on the experimented datasets).

**Results.**

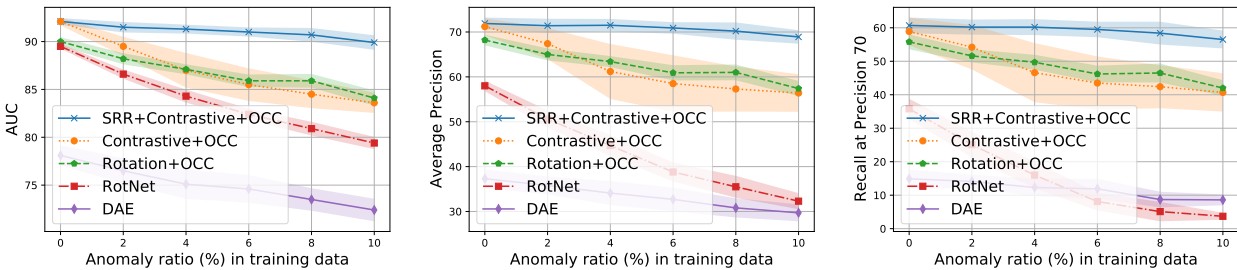

Figure 5: Unsupervised AD performance with various OCCs on CIFAR-10 dataset. For SRR we adapt distribution-augmented contrastive representation learning (Sohn et al., 2021). (Left) AUC, (Middle) Average Precision (AP), (Right) Recall at Precision 70.

On CIFAR-10 dataset, Fig. 5 compares SRR to other alternatives. In general, most methods suffer from a significant performance drop with increased anomaly ratio, despite using representation learning. For example, the AUC for AD based on distribution-augmented contrastive representation (Sohn et al., 2021) drops from 92.1 to 83.6 when anomaly ratio becomes 10%. Similarly, with the improved rotation prediction representation (Komodakis & Gidaris, 2018), the AD performance drops from 90.0 to 84.1 in AUC. On the other hand, SRR effectively handles the contamination in training data and the reduction in performance is much lower with increased anomaly ratio – it achieves 89.9 AUC with 10% anomaly ratio, reducing the performance drop by 74.1% compared to the best alternative. One 'oracle' upper bound would be removal of all anomalies from training data, which is the same as the performance at 0% anomaly ratio for the same size of the data. As Fig. 5 shows, the performance of SRR is similar to this oracle upper bound, with less than 2.5 AUC difference, up until high anomaly ratios (10%). The results are also similar in other metrics, such as AP and Recall at Precision of 70, described in Fig. 5.

Next, we show additional results on 3 visual AD datasets in Fig. 6, comparing to the best alternatives with representation learning and OCC. The improvements of SRR, particularly at higher anomaly ratios, are consistent across different cases. For example, on MVTec dataset, SRR improves AUC by 4.9 and AP by 7.1 compared to the state-of-the-art CutPaste OCC with an anomaly ratio of 10%. On Dog-vs-Cat task, we observe the improvement to be smaller, which we attribute to similarity of the learned representation for these two classes, that makes anomaly definition more ambiguous.

## 5  Performance Analyses

In this section, we focus on providing more insights on the important constituents of our framework with empirical results. We first present ablation studies for better understanding of the source of gains with different components of the framework. Next, we quantify the refinement accuracy and show the efficacy of the proposed refinement block, as one of the key components of the proposed framework. Finally, we present sensitivity analyses on two key hyperparameters of SRR, as it is crucial for an unsupervised AD method not to be sensitive to its hyperparameters.

### 5.1  Ablation studies

To shed more light on the source of gain coming from different components, we present ablation studies. Specifically, we include comparisons with the final ensemble model on the converged self-supervised extractor (SRR without the final OCC), without using data refinement but only employing an ensemble of OCCs, applying data refinement only for OCCs (but not for the representation updates), and data refinement with majority voting (SRR with majority voting).

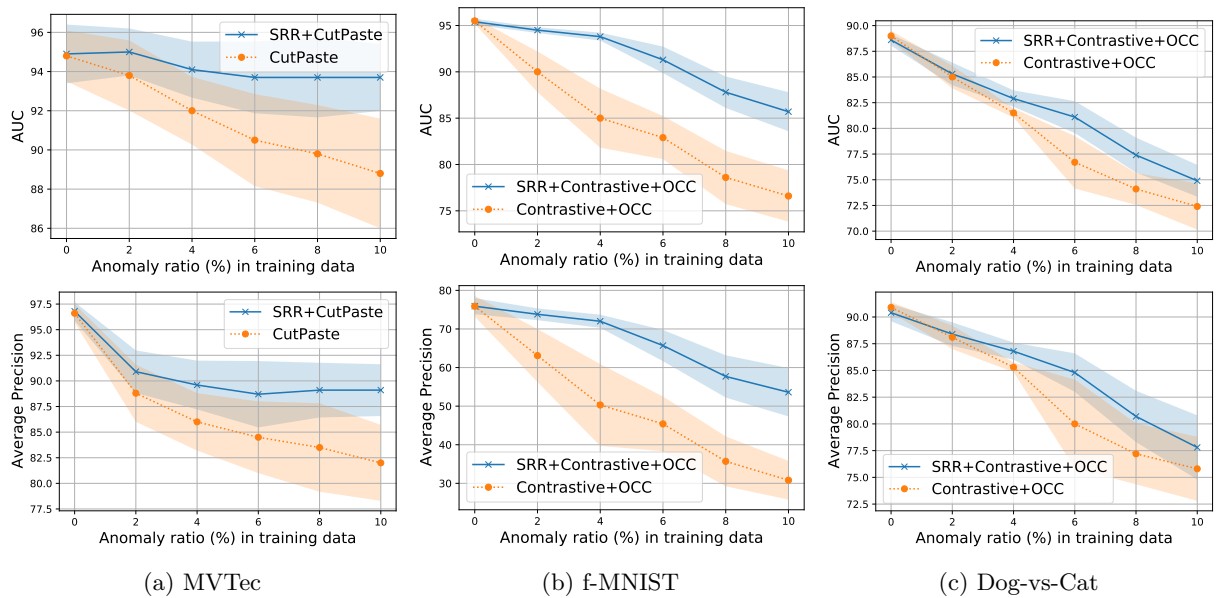

(a) MVTec  (b) f-MNIST  (c) Dog-vs-Cat

Figure 6: Unsupervised AD performance on (a) MVTec (b) f-MNIST, and (c) Dog-vs-Cat datasets with varying anomaly ratios. We use state-of-the-art one-class classification models for baselines, such as distribution-augmented contrastive representations (Sohn et al., 2021) for f-MNIST and Dog-vs-Cat, or CutPaste (Li et al., 2021) for MVTec, and build SRR on top of them.

Table 1: Ablation studies. SOTA OCC methods are CutPaste (Li et al., 2021) for MVTec dataset and (Sohn et al., 2021) for CIFAR-10 dataset with 6% noise. Metrics are (AUC/AP).

| Methods / Datasets | MVTec | CIFAR-10 |
|---|---|---|
| SOTA OCC | $0.905\pm0.024$ / $0.845\pm0.035$ | $0.855\pm0.017$ / $0.585\pm0.063$ |
| SRR without final OCC | $0.922\pm0.022$ / $0.870\pm0.032$ | $0.890\pm0.007$ / $0.677\pm0.019$ |
| SRR without data refinement (Ensemble of OCCs) | $0.911\pm0.023$ / $0.849\pm0.037$ | $0.862\pm0.010$ / $0.599\pm0.031$ |
| Data refinement for only OCC | $0.918\pm0.028$ / $0.858\pm0.039$ | $0.885\pm0.006$ / $0.644\pm0.016$ |
| SRR with majority voting | $0.925\pm0.017$ / $0.873\pm0.026$ | $0.893\pm0.005$ / $0.675\pm0.014$ |
| SRR | $\mathbf{0.937\pm0.018}$ / $\mathbf{0.887\pm0.032}$ | $\mathbf{0.910\pm0.005}$ / $\mathbf{0.709\pm0.013}$ |

In Table. 1, the proposed version of SRR (with an additional final OCC) outperforms the version without the final OCC. Because while fitting the individual OCC models in the ensemble, we do not exclude the possible anomaly samples for diversity of the trained submodels, so the anomaly decision boundaries can be fitted robustly. Ensemble of one-class classifiers is employed to identify the possible anomalous samples in the training set rather than yielding final anomaly score predictions. Therefore, we do not exclude the possible anomalous samples to train the weak one-class classifier. In that regard, if we directly utilized the outputs of the ensemble for the final anomaly scores, the performance would be worse (as shown in Table 1). Also, employment of ensemble of OCCs without data refinement, yields worse performance than the proposed (SRR), underlining the importance of the core data refinement idea of SRR, which was further eluded in Sec. 5.2. Lastly, as opposed to applying data refinement only for OCCs, we find that learning representation with refined data plays a crucial role, resulting in another improvement compared to SRR using fixed representations trained on the entire dataset. The proposed version of SRR achieved statistically significant improvements over alternatives with CIFAR-10 datasets in terms of both AUC and AP but not with MVTec datasets due to their small sample size per category (100s-500s).

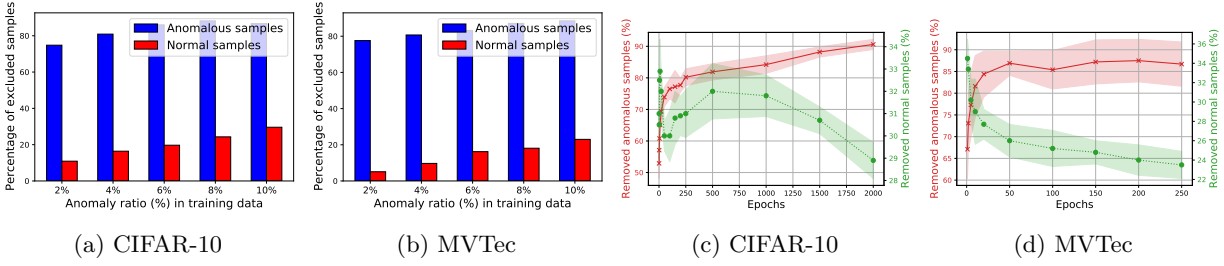

(a) CIFAR-10      (b) MVTec      (c) CIFAR-10      (d) MVTec

Figure 7: Percentage of excluded anomalous and normal samples by data refinement (a, b) with different anomaly ratios in training data and (c, d) over training epochs with 10% anomaly ratio.

## 5.2 Refinement efficacy

It is crucial for the proposed data refinement block being accurate in filtering the correct anomaly samples, to minimize the impact of contamination. We evaluate the accuracy of this filtering of the proposed data refinement block. As Fig. 7(a, b) shows, with data refinement, we can exclude more than 80% of anomalies in the training set without removing too many normal samples. For example, among 4% anomalies in CIFAR-10 data, SRR is able to exclude 80% anomalies while removing less than 20% normal samples. Such a high recall of anomalies of SRR is not only useful for unsupervised AD, but can also help improving the annotation efficiency when a budget for active learning is available. Fig. 7(c, d) demonstrates the removed normal and abnormal samples by the data refinement module over training epochs. It shows that better representation learning (as the number of training epochs increases) consistently improves the efficacy of the data refinement. This positive reinforcement between better representation learning and higher filtering accuracy, constitute one key aspect of the proposed framework.

## 5.3 Sensitivity to hyperparameters

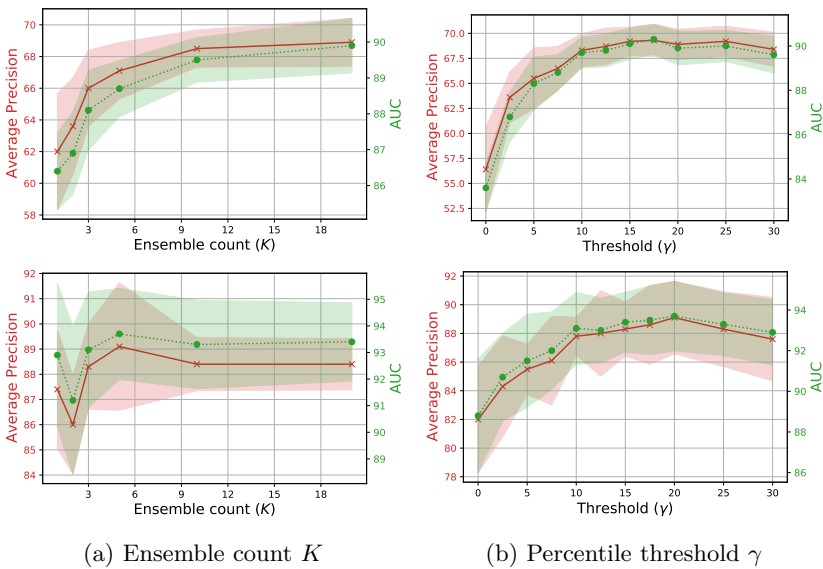

(a) Ensemble count $K$      (b) Percentile threshold $\gamma$

Figure 8: On (top) CIFAR-10 and (bottom) MVTec under 10% anomaly ratio setting, sensitivity analyses for the (a) ensemble count $K$, (b) percentile threshold $\gamma$.

SRR is designed for fully-unsupervised AD, thus, it is important to ensure robust performance against changes in the hyperparameters, as model selection without labeled data would be very challenging. To demonstrate this, we conduct sensitivity analyses for the key hyperparameters, shown in Fig. 8.

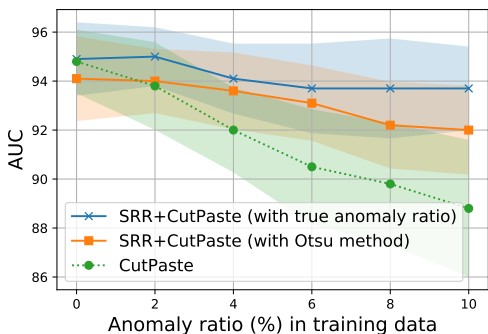

Figure 9: Unsupervised AD performance for SRR with Otsu's method on MVTec dataset in AUC.

Fig. 8a shows the impact of the number of OCCs in the ensemble, $K$. In general, the performance saturates even after a small number of OCCs, and a typical value of $K$ around $\sim$10 seems applicable across most datasets. On CIFAR-10 (top in Fig. 8a), we observe slight improvements with higher $K$ as the number of samples to train each OCC ($N/K$) would be sufficient.

Fig. 8b, shows the impact of the threshold parameter, $\gamma$. Overall, an intermediate value seems optimal and the performance does not change much around it. We observe that SRR performs robustly when $\gamma$ is set to be larger than the actual anomaly ratio (10%). When $\gamma$ is less than 10%, however, we observe some drop in performance as $\gamma$ decreases. Yet, we note that SRR still improves over alternatives regardless of the threshold. The analyses suggest that $\gamma$ could be set to be anywhere from the true anomaly ratio and and $2\times$ the anomaly ratio to maximize its effectiveness. For the scenario of unknown true anomaly ratio, we propose Otsu's method (see Sec. 6).

## 6   SRR with unknown anomaly ratio: Otsu's method

Proposed AD framework (SRR) is significantly better than alternatives in fully-unsupervised setting, as shown in wide range of scenarios. Despite being in fully-unsupervised setting, one information reliance of SRR is the true anomaly ratio. In practice, many applications have a good estimate for it from domain knowledge – e.g. banks know typical credit card fraud ratio. Yet, there could be some applications that the true anomaly ratio may also be completely unknown. In this section, we propose an extension of SRR to address this scenario – performing unsupervised AD without any information on anomaly ratio.

A key component would be automatic threshold selection. Some previous works (Xia et al., 2015; Beggel et al., 2019; Pang et al., 2020) studied metrics like intra-class variance minimization for determining the threshold for data refinement. In a similar vein, we propose integrating Otsu's method (Sezgin & Sankur, 2004) into SRR to identify the threshold between normal and anomalous samples. We use this threshold for selecting the hyper-parameter ($\gamma$) instead of twice of the true anomaly ratio. The key idea of the Otsu's method is aiming to find the threshold that minimizes the intra-class variance. This is defined as the weighted sum of variances of the two classes [4]. Let us denote the normality scores as $\{s_i\}_{i=1}^N$ and threshold as $\eta$. Then, we pick the threshold ($\eta$) that minimizes the weighted sum of the variance ($w_0(\eta) \times \sigma_0(\eta) + w_1(\eta) \times \sigma_1(\eta)$) where $w_0(\eta) = \sum_{i=1}^N \mathbb{I}(s_i < \eta)/N$ and $w_1(\eta) = \sum_{i=1}^N \mathbb{I}(s_i \geq \eta)/N$. $\sigma_0(\eta)$ and $\sigma_1(\eta)$ are the variances of each class. The optimal threshold ($\eta^*$) is determined as

$$\eta^* = \min_{\eta} w_0(\eta) \times \sigma_0(\eta) + w_1(\eta) \times \sigma_1(\eta). \tag{3}$$

We use the twice of $\eta^*$ as the hyperparameter ($\gamma$) in SRR based on the sensitivity analyses described in Sec. 5.3.

We evaluate the performance of SRR with Otsu's method and compare to the state-of-the-art OCC (Li et al., 2021) and the original SRR with the knowledge of the true anomaly ratio on MVTec dataset. Fig. 9

---

[4]More details can be found here `https://en.wikipedia.org/wiki/Otsu%27s_method`

Table 2: Performances with Otsu's method. SOTA OCC methods are (Sohn et al., 2021) for CIFAR-10 dataset and GOAD (Bergman & Hoshen, 2019) for Thyroid dataset. We introduce 6% noise on MVTec dataset and 1.5% noise on the Thyroid dataset. Metrics are (AUC/AP) for CIFAR-10 dataset and F1 score for Thyroid dataset.

| Methods / Datasets | CIFAR-10 | Thyroid |
|---|---|---|
| SOTA OCC | 0.855 / 0.585 | 0.506 |
| SRR with Otsu's method | 0.906 / 0.703 | 0.623 |
| SRR | **0.910 / 0.709** | **0.639** |

demonstrates that even without true anomaly ratio, the performance of SRR can be significantly better than the state-of-the-art OCC (Li et al., 2021) with Otsu's method. Per results, the knowledge of true anomaly ratio is definitely a useful information for maximizing the performance of SRR in fully unsupervised settings. We further extend the experimental results using Otsu's method with other datasets such as CIFAR-10 and Thyroid. Similarly, Table 2 show that Otsu's method yields only slight degradation compared to SRR with true anomaly ratio; however, it still significantly outperforms SOTA OCC baselines.

## 7 Conclusion

AD has wide range of use cases with significant importance in real world applications, from detecting security threats to the financial system to identifying faulty behaviors of manufacturing machines. A challenging and costly aspect of building an AD system is that anomalies are rare and not easily detectable by humans, yielding high complexity and cost. To this end, we propose, SRR, a canonical AD framework to enable high performance AD without any labels. SRR employs an ensemble of multiple OCCs to propose candidate anomaly samples that are refined from training, which allows more robust fitting of the anomaly decision boundaries as well as better learning of data representations. SRR can be flexibly integrated with any OCC, and applied on raw data or on trainable representations. We demonstrate the state-of-the-art AD performance of SRR on multiple tabular and image datasets from various applications. We provide detailed analyses on the key contributing factors of SRR, which we hope to provide further guidance in AD research. Lastly, we extend SRR to the scenario of not possessing any information on the anomaly ratio. We leave some important aspects, explainability and reliability of AD, to future work.

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

# A   Additional results

## A.1   SRR on raw tabular features / learned image representations

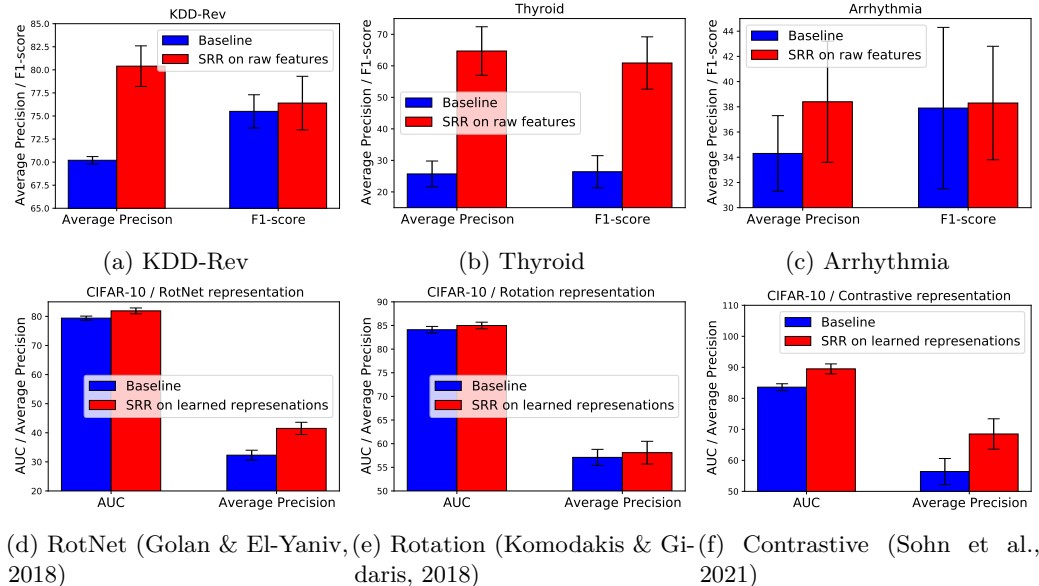

(a) KDD-Rev          (b) Thyroid          (c) Arrhythmia

(d) RotNet (Golan & El-Yaniv, 2018)    (e) Rotation (Komodakis & Gidaris, 2018)    (f) Contrastive (Sohn et al., 2021)

Figure 10: Performance of SRR on (top) raw tabular features and (lower) learned image representations. SRR consistently outperforms baseline and in some cases (e.g., Thyroid, and Contrastive (Sohn et al., 2021)), the performance improvements are significant.

SRR is also applicable on raw tabular features or learned image representation without representation update using data refinement. In this section, we demonstrate the performance improvements by SRR without representation update to verify the effectiveness of data refinement block of SRR for shallow OCCs.

Fig. 10 (upper) demonstrates consistent and significant performance improvements when we apply SRR on top of raw tabular features. Specifically, the Average Precision (AP) improvements are 10.2, 29.0, and 4.1 with KDD-Rev, Thyroid, and Arrhythmia tabular datasets, respectively. We also apply SRR on top of various learned image representations. As can be seen in Fig. 10 (lower), the performance improvements of SRR are consistent across various different learned image representations (without representation update). For instance, the AP improvements are 9.2, 1.0, and 12.1 with learned image representations using RotNet (Golan & El-Yaniv, 2018), Rotation (Komodakis & Gidaris, 2018), and Contrastive (Sohn et al., 2021), respectively.

## A.2   Additional baselines

Table 3: Additional experiments with extra baselines from robust AD literature. We introduce 6% noise on CIFAR-10 and KDD datasets. For Thyroid dataset, we introduce 1.5% noise. Metrics are (AUC/AP) for image data and F1 score for tabular data.

| Methods / Datasets | CIFAR-10 | Thyroid | KDD |
|---|---|---|---|
| PCA | - | 0.299 | 0.836 |
| Robust PCA | - | 0.377 | 0.893 |
| LOF | - | 0.338 | 0.873 |
| Robust AE | 0.636 / 0.174 | - | - |
| SRR | **0.910 / 0.709** | **0.506** | **0.942** |

We add extra baselines from robust AD literature: Standard PCA (Wold et al., 1987), Robust PCA (Candès et al., 2011) and Local Outlier Factor (LOF) (Breunig et al., 2000) for tabular data and Robust autoencoder (Robust AE) (Zhou & Paffenroth, 2017) for image data. In Table 3, the performance of PCA and LOF are highly degraded even with a small amount of anomalies in the training data. For Robust PCA and Robust AE, the performance degradation is less but still significant in comparison to SRR. Overall, SRR outperforms other benchmarks in fully unsupervised settings, underlining the importance of data refinement in improving the robustness with contaminated data, as the core constituent of SRR.

### A.3 Convergence graphs

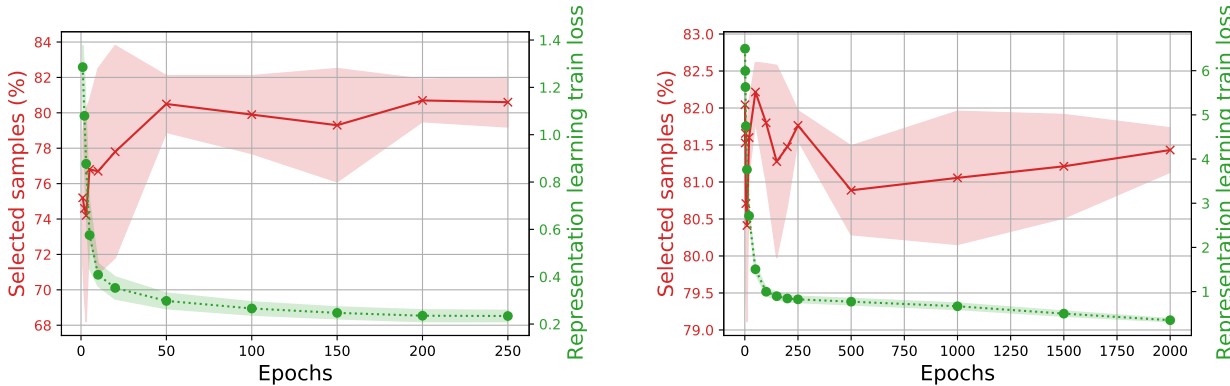

Figure 11: Convergence graphs of SRR with (left) MVTec dataset, (right) CIFAR-10 dataset.

Convergence is a well-known challenge in alternating training. It is more problematic in unsupervised learning settings due to the lack of validation set to evaluate the performance convergence. In the proposed framework, feature extractor loss is used for the convergence criteria (if no improvement is observed in the loss for 5 epochs) and data refinement block is intermittently updated with much lower frequency. The data refinement block consists of shallow one-class classifiers (not neural networks) trained on top of learned representations. Empirically, when the feature extractor's loss converges, the data refinement block is also converged. Fig. 11 illustrate the convergence graphs of SRR with MVTec and CIFAR-10 datasets.

## B Computational complexity

Note that when applying SRR on top of representation learning, the computational complexity of the representation learning part is not changed. With SRR, the additional computations come from training the ensemble models on top of learned representations. Note that we use shallow one-class classifiers (such as GDE or one-class SVM) for submodels; thus, the additional computational complexity is marginal. We would like to mention that all the experiments are done on a single V100 GPU and each experiment needs at most 12 hours for training (the additional training time caused by the SRR framework is an average 13.1% of the total training time). The computations of the ensemble parts can be further improved by the model parallelization.

## C Implementation details for GOAD (Bergman & Hoshen, 2019)

A classification-based AD method, GOAD (Bergman & Hoshen, 2019), has demonstrated strong AD performance on tabular datasets. Unlike previous works (Golan & El-Yaniv, 2018; Hendrycks et al., 2018) that formulate a parametric classifier for multiple transformation classification, GOAD employs distance-based classification of multiple transformations. For the set of transformations $T_m : \mathcal{X} \to \mathcal{D}$, $m = 1, ..., M$, the loss

function of GOAD is written as in Eq. 4 with the probability defined in Eq. 5.

$$\mathcal{L} = -\mathbb{E}_{m,x}\big[\log P(m|T_m(x))\big], \tag{4}$$

$$P(\hat{m}|T_m(x)) = \frac{\exp(-\|f(T_m(x)) - c_{\hat{m}}\|^2)}{\sum_n \exp(-\|f(T_m(x)) - c_n\|^2)}, \tag{5}$$

where the centers $c_m$'s are updated by the average feature over the training set. While it is shown to perform well (Bergman & Hoshen, 2019), we find that the distance-based formulation is not necessary, and we achieve the similar performance, if not worse, to (Bergman & Hoshen, 2019) using a parametric classifier when computing the probability:

$$P(\hat{m}|T_m(x)) = \frac{\exp\big(w_{\hat{m}}^\top f(T_m(x)) + b_{\hat{m}}\big)}{\sum_n \exp\big(w_n^\top f(T_m(x)) + b_n\big)} \tag{6}$$

The formulation in Eq. 6 is easier to optimize than its original form in Eq. 5 as it can be fully optimized with backpropagation without alternating updates of feature extractor $f$ and centers $c_m$. Once we learn a representation by optimizing the loss in Eq. 4 using Eq. 6, we follow a two-stage one-class classification framework of (Sohn et al., 2021) to construct a set of Gaussian density estimation OCCs for each transformation. Finally, we aggregate a maximum normality scores from a set of classifiers as the normality score.

In Table 4, we summarize the implementation details, such as network architecture or hyperparameters, and AD performance under clean training data setting that reproduces the results in (Bergman & Hoshen, 2019).

Table 4: The AD performance under clean only data setting of GOAD in (Bergman & Hoshen, 2019) and our implementation. Our implementation demonstrates comparable, if not worse, performance to those reported in (Bergman & Hoshen, 2019). Our implementation also shares most hyperparameters across datasets except the $M$, the number of transformations, and the train steps, which are closely related to the size of training data.

| Datasets | KDD | KDD-Rev | Thyroid | Arrhythmia |
|---|---|---|---|---|
| F-score (Bergman & Hoshen, 2019) | $98.4_{\pm 0.2}$ | $98.9_{\pm 0.3}$ | $74.5_{\pm 1.1}$ | $52.0_{\pm 2.3}$ |
| F-score (ours) | $98.0_{\pm 0.2}$ | $95.0_{\pm 0.2}$ | $75.1_{\pm 2.4}$ | $54.8_{\pm 3.2}$ |
| $f$ (feature) | $\big[\text{Linear(8), LeakyReLU(0.2)}\big] \times 5$ | | | |
| Optimizer | Momentum SGD (momentum$=0.9$) | | | |
| Learning rate | 0.001 | | | |
| Batch size | $64 \times M$ | | | |
| L2 weight regularization | 0.00003 | | | |
| Random projection dimension | 32 | | | |
| $M$ | 32 | | 256 | |
| Train steps | $2^{10}$ | | $2^{16}$ | |

