# OpenReview forum: "Self-supervise, Refine, Repeat: Improving Unsupervised Anomaly Detection"
_TMLR — Accepted by TMLR_

### Review · Reviewer_CDcs · 2022-06-23

**Summary Of Contributions:**

This paper studied the unsupervised anomaly detection problem and proposed a method based on self-supervised learning, an ensemble of one-class classifiers, and threshold-based sample selection.
The proposed method was validated empirically on the semi-synthetic image (CIFAR-10, Fashion MNIST, Asirra, MVTec AD) and tabular datasets.

Concretely, the proposed method, referred to as "self-supervise, refine, repeat" (SRR), first partitions the training set of normal and abnormal instances into subsets and trains multiple one-class classifiers;
Then, the abnormal training instances are detected based on a predefined and fixed thresholding hyperparameter; Meanwhile, the feature extractor is trained only on selected normal training instances via self-supervised learning; Finally, a single one-class classifier is trained on selected normal training instances.


**Broader Impact Concerns:**

I do not foresee such concerns.

**Requested Changes:**

- What are "adversarial outcomes at intensive care units"?
- "They all depend on some labeled data", and "Most prior unsupervised AD works assume that the data contain entirely negative samples"... are strawman arguments and not justifiable.
- Vacuous intensifiers like "impressive performance", "quite sub-optimal", "holistic approach," and "fundamental challenges" are not scientific writing and are debatable.
- In my opinion, this work is not data-centric: there is only one application -- anomaly detection, and the main contribution is still model and training method design.
- Either all "percentile" need to be replaced by "quantile", or $\gamma$ in Eq. (2) needs to be multiplied by $100$.
- "We proposed that it is beneficial to..." sounds unnatural.
- The criterion of convergence is not given.
- Figure 4: F-score?
- Otsu's method: why weighted?


**Strengths And Weaknesses:**

## Strengths

1. Unsupervised anomaly detection is a challenging task.
2. The author provided empirical evidence that this training scheme based on ensemble and self-supervised learning may be helpful.
3. The author studied the problem of unknown anomaly ratios and introduced a method for such a case.

## Weaknesses

The biggest problem is that, for almost all parts, the author only stated what they did but not why they did so. Since all components are not novel (self-supervised learning, one-class classifier, ensemble, etc.), it is crucial that all design choices are supported by a reasonable explanation, theoretical justification, or empirical evidence. For example:

- The author proposed to use an ensemble of one-class classifiers. It is well known that bagging could improve the expressiveness of models and enhance weak learners. But what if we only use a single classifier and train it along with the feature extractor? What will the problem be? In other words, why is an ensemble preferable or necessary? If we choose ensemble, why do we choose unanimity voting over other methods such as majority voting? These issues were not well explained, referred to a reliable source, or empirically supported.
- The author chose to train the feature extractor and one-class classifiers simultaneously. However, the benefit is unclear. Since the number of positive instances is small (by definition of anomaly detection), it seems no problem to train a feature extractor with self-supervised learning first with all training instances. We can even use pre-trained ones if available. The author did show some empirical evidence from an ablation study, which is, however, based on a single setting and a single trial. The statistical significance may not be high enough.
- Another design choice is the last one-class classifier. The explanation "the proposed version of SRR (with an additional final OCC) outperforms the version with the final OCC, which is attributed to the fact that while fitting the individual OCC models in the ensemble, we do not exclude the possible anomaly samples for diversity of the trained submodels, so the anomaly decision boundaries can be fitted robustly" is not understandable.

Another issue is the experimental setting based on semi-synthetic datasets. It is questionable if treating one class as normal and others as abnormal from the CIFAR-10, Fashion MNIST, and Asirra datasets reflects actual anomaly detection applications in the real world.

There are other expression issues, strawman arguments, and typos, which are listed below.

---

> ### Author Response · Authors · 2022-07-08
> **RE: Review of Paper187 by Reviewer CDcs**
>
> Thank you for your valuable comments on our work. Hopefully, our responses will help resolve your concerns.
>
> **Answer 1**: We agree that the benefits of bagging ensembles to improve the performance of weak learners are well-known, particularly for supervised learning settings. Regarding our method, indeed ensemble is very helpful to improve the anomaly detection performance and data refinement in unsupervised settings. We have already done extensive analysis on this and presented the results in Fig. 8. As can be seen in Fig 8(a), the performance is much better for higher ensemble counts, compared to a single classifier training along with the feature extractor (0.065 / 0.017 AP improvements with CIFAR-10 / MVTec datasets). The improvement saturates beyond some value, typically around ~10 classifiers. We attribute the benefit of ensembling to improved robustness against OCC overfitting and hence superior data refinement. We have added a clarification sentence in Sec. 3.1.
>
> **Answer 2**: Thanks for suggesting this idea. As explained in Section 3.1, we claim that it is critical to exclude true-anomalous samples from the training set, as much as possible. Unanimity voting is a more aggressive method in removing the possible anomalous samples from the training set compared to the majority voting; thus, we select the unanimity voting rather than majority voting. As can be seen in Fig 8(b), more aggressive methods for removing the possible anomalous samples (higher threshold) show better performance. We also included the experimental results with majority voting in Table 1 in the revised manuscript - unanimity voting achieves 0.012 / 0.014 and 0.017 / 0.034 better AUC / AP than majority voting for MVTec and CIFAR-10 datasets, respectively.
>
> **Answer 3**: First, we show the harm of a small number of positive instances in the training set for anomaly detection in Fig 2 - they deteriorate the performance significantly. This is also valid for other self-supervised learning based models such as RotNet and DAE (in Fig 5). Thus, it would be problematic to train a feature extractor with all training samples with anomalous samples in the training set. It is critical to apply self-supervised learning with data refinement and then train the one-class classifiers on the refined data.
>
> Regarding statistical significance of the results, all the reported results are averaged over 5 independent runs. We have included the average standard deviations in Table 1 (Ablation studies) in the revised manuscript. It shows that the AUC and AP improvements are statistically significant on the CIFAR-10 dataset. On the MVTec dataset, the improvements are not statistically significant due to the small number of samples (100s-500s samples per category); however, the absolute improvements are on average 0.021 / 0.028 in terms of AUC / AP. We have added a note on this in the revised manuscript.
>
> **Answer 4**: Thanks for raising this concern. We have clarified the explanation as:
>
> “The proposed version of SRR (with an additional final OCC) outperforms the version without the final OCC. Because while fitting the individual OCC models in the ensemble, we do not exclude the possible anomaly samples for diversity of the trained submodels, so the anomaly decision boundaries can be fitted robustly. Ensemble of one-class classifiers is employed to identify the possible anomalous samples in the training set rather than yielding final anomaly score predictions. Therefore, we do not exclude the possible anomalous samples to train the weak one-class classifier. In that regard, if we directly utilized the outputs of the ensemble for the final anomaly scores, the performance would be worse (as shown in Table 1).”
>
> **Answer 5**: Note that many other anomaly detection papers utilize those datasets as well, to show the anomaly detection performances (please check this link for various papers: https://paperswithcode.com/task/anomaly-detection). We also use the MVTec AD dataset and 4 different tabular datasets which are based on the real-world anomaly detection applications.
>
> **Answer 6**: Death, heart attack, and blood poisoning can be good examples of adversarial outcomes at ICUs - we have clarified.
>
> **Answer 7**: In the revised manuscript, we have made our claims more precise. Based on our literature review (please see our introduction and related work sections), most papers in the anomaly detection field rely on some labeled data and it is hard to find the prior works which are designed for fully-unsupervised anomaly detection.
>
> **Answer 8**: We have removed these expressions to make our language more scientifically rigorous.

---

> > ### Author Response · Authors · 2022-07-08
> > **RE2: Review of Paper187 by Reviewer CDcs**
> >
> > **Answer 9**: The terminology “data-centric” is often used for methodologies where the focus is on the usage of data, rather than judicious model improvements [https://spectrum.ieee.org/andrew-ng-data-centric-ai]. We have added a footnote to clarify this.  Our contributions are mainly about refining data from the unlabeled data instead of improving the model without modifying the usage of data. In that point of view, we claim that the proposed method can be interpreted as the data-centric approach. A data-centric method does not need to be application-agnostic, it can be for supervised learning or anomaly detection.
> >
> > **Answer 10**: Thank you for raising this typo. We have added 100 multiplications in Eq. (2).
> >
> > **Answer 11**: We have revised it as “we empirically showed that it is beneficial to …”
> >
> > **Answer 12**: If the training loss is converged (if no improvement is observed in the loss for 5 epochs), the models are converged as well. We have clarified this in the revised manuscript.
> >
> > **Answer 13**: Thank you. We have fixed them as the F1-scores in the revised manuscript.
> >
> > **Answer 14**: The original Otsu’s method is defined as minimizing the intra-class variance using weighted variance sum. Please see this link for more information about Otsu’s method (https://en.wikipedia.org/wiki/Otsu%27s_method). We have added a note to clarify this.

---

### Review · Reviewer_GJKz · 2022-06-27

**Summary Of Contributions:**

This paper proposes a self-supervised ensemble method for unsupervised anomaly detection. In contrast to the previous unsupervised model-centric method, the proposed method is data-centricc, and this allows more robust fitting of the anomaly decision boundaries and also better learning of data representations. Experiments on tabular and image datasets validate that the proposed  approach is more robust at high anomaly contamination ratios than respective state-of-the-art single detectors.

**Requested Changes:**

As I listed in the weaknesses, it would be nice if the authors could further list their contributions in the introduction. Moreover, I think the authors should slightly reduce the discussion of the unsupervised problem setting.

**Strengths And Weaknesses:**

**Strengths**:

1). Overall, this paper is well-written, and very easy to follow, even though I am not very familiar with the anomaly detection task.

2).The idea is simple and natural. Using (multiple) one-class classiers for anomaly detection seems technically sound to me. There are K detectors together with a joint self-supervised feature extractor g on K disjoint subsets of the data. The final detector is an integration of the K detectors, so that the final performance is improved.

3). The effectiveness of the proposed framework is validated on top of contrastive learning-based models, which are state-of-the-art. Extensive experiments on both tabular and image datasets validate the effectiveness and superiority of the framework. Ablation studies well isolate the effects of each hyperparameter.


**Weaknesses**:

1). Although the idea is simple and natural, the methodological novelty is somewhat limited. Actually, it is well known that emsemble methods can improve the model robustness in many recognition and detection tasks.

2). The real contributions of this paper should be further clarified. This is not the first work that focuses on the problem setting of unsupervised anomaly detection, but in the abstract and Fig. 1, I think the authors discuss too much about the concept of this problem setting.

3). As we know, the iterations of K detectors and feature extractors and alternative. How can we ensure that this process will certainly converge? It would be nice if the authors could provide some theoretical discussions or insights.


**In Summary**:

Overall, although the technical novelty is not too high, I still think this is a good paper and I would like to give an accept for it. If it is finally accepted, please consider my questions listed in the weakness points as well as the suggestions in the requested changes.

---

> ### Author Response · Authors · 2022-07-08
> **RE: Review of Paper187 by Reviewer GJKz**
>
> Thank you for your valuable comments on our work.
>
> **Answer 1**: We acknowledge that the ensemble method itself is not a novel idea. However, we note that the proposed method is not just applying ensemble learning to the unsupervised anomaly detection. We utilize the ensemble method to identify the potential positive samples from the unlabeled dataset as a data-centric approach. At the end, the final one-class classifier is a single classifier and not the ensemble classifiers which reflect that the performance improvements are not just based on the well-known ensemble techniques.
>
> **Answer 2**: Thank you for raising this point. We have modified our expression to clarify that our paper is not the first paper to study unsupervised anomaly detection, but rather most (not all) papers indeed do not focus on the truly-unsupervised setting, as they assume the training data contain only negative samples (which inherently has the label knowledge). We have also clearly listed the contributions in the Introduction in the revised manuscript (see **Answer 4** for more details).
>
> **Answer 3**: Convergence is a well-known challenge in alternating training. It is more problematic in unsupervised learning settings due to the lack of validation set to evaluate the performance convergence. In the manuscript, we present the convergence graphs in Appendix A.3 and show that the feature extractors (representation learners) losses converge in a stable way. To achieve stable convergence of this alternative training procedure in the proposed method, we only update the K detectors (one-class classifiers) once per multiple epoch (see Implementation details in Section 4).
>
> **Answer 4**: Thank you for the great suggestions. We have added the below listed contributions in the revised manuscript.
>
> - We propose a novel data-centric framework, SRR, for unsupervised anomaly detection using the ensemble of one-class classifiers as a data refinement module.
> - The proposed framework is a model-agnostic approach that can be applicable on top of any anomaly detection framework, considering different ways of applying self-supervised representation learning or one-class classification.
> - SRR achieves significant robustness improvements with various anomaly ratios - in other words, the users do not need to worry about manually filtering the possible anomalies from the training data to minimize contamination. We demonstrate superior performances on multiple tabular and image datasets.

---

### Review · Reviewer_Gw8t · 2022-07-04

**Summary Of Contributions:**

This paper studies the unsupervised anomaly detection problem where the unlabeled training data contains an unknown portion of anomalies. As the anomalies in training data degrade the performance of one-class classification based on unsupervised anomaly detection methods, they propose an ensemble method that trains multiple one-class classifiers on disjoint subsets of training data, uses them for detecting potential anomalies, and then excludes the detected anomalies from the training data. They iteratively conduct the ensemble training and the training of a feature extractor on the filtered training data using self-supervised objectives until its convergence. Lastly, a one-class classifier is trained on the finally refined data. Experiments on various datasets and settings demonstrate the effectiveness of the proposed method.


**Broader Impact Concerns:**

-The proposed approach seems more general than the given algorithm. It does not depend on the ensemble training for data refinement or the self-supervised training for learning representations. Other frameworks can also be applied, and it would help make the work more impactful if such discussions can be provided.

**Requested Changes:**

-It would interesting to see discussions in what situations they cannot converge. In section A.3 some convergence graphs are provided. It would be helpful if more insights and discussions on why they always converge in experiments can be given.

-As the proposed method is based on ensemble models and iterative training, comparing the computational time and cost with baseline methods may also be needed.

-In table 1, it would be helpful if the standard deviation results can also be reported.




**Strengths And Weaknesses:**

Pros:

-The paper is clear and easy to follow.

-The paper includes thorough experimental results with ablation studies and sensitivity analysis on hyperparameters. It is interesting to see the performance improvement brought by retraining the representations using refined training data.

Cons:

-The proposed method iteratively trains a distiller that excludes possible anomalies based on ensemble models and a feature extractor based on self-contrastive learning using the distilled data. Such an ensemble training nature may bring heavy computation costs.

-Another concern is that the convergence of the distiller and the feature extractor in the iterative training procedures may not be guaranteed.

---

> ### Author Response · Authors · 2022-07-08
> **RE: Review of Paper187 by Reviewer Gw8t**
>
> Thank you for your valuable comments on our work.
>
> **Answer 1**: Thank you for raising this concern. We agree on the importance of computational complexity and we have already included some analyses in the manuscript. More details can be found in Appendix B and at the beginning of Section 4.
>
> In Appendix B, we quantitatively compare the computational complexity with the baseline; SRR only shows 13.1% training time increase compared to the baseline. We note that this small difference is due to our judicious design. First, in order to avoid the significant increase of the computations, we use shallow one-class classifiers (such as GDE) as the weak learners of the ensemble models. Also, we only update the data refinement block at certain epochs (e.g., after 500 epochs, we only update the data refinement block per each 500th epoch). Therefore, the additional computations with the proposed method are marginal.
>
> **Answer 2**: We acknowledge that the proposed method does not guarantee the convergence of the distiller and the feature extractor; and this convergence problem is well-known for the alternating training frameworks. In the proposed framework, feature extractor loss is used for the convergence criteria (if no improvement is observed in the loss for 5 epochs) and data refinement block is intermittently updated with much lower frequency. The data refinement block consists of shallow one-class classifiers (not neural networks) trained on top of learned representations. With the converged feature extractor, the corresponding data refinement block would also be converged. We have added sentences to further explain this in the revised appendix.
>
> **Answer 3**: We have added the standard deviations results in Table 1 in the revised manuscript.
>
> **Answer 4**: Thank you for raising this point. SRR is a data-agnostic and model-agnostic approach that can be applicable to different types of data and one-class classification models. To show this generalizability, we demonstrated the experimental results with multiple tabular datasets (KDD, KDD-Rev, Thyroid, Arrhythmia) and image datasets (CIFAR-10, MVTeC, Dog-vs-Cat, f-MNIST) - see Section 4. Also, we applied the proposed SRR on top of multiple different one-class classification models (GOAD for 4 tabular datasets, Contrastive learning for CIFAR-10, f-MNIST, Dog-vs-Cat, and CutPaste for MVTeC). We clarified this generalizability of the proposed method in the introduction in the revised manuscript.

---

### Comment · Action_Editors · 2022-07-19
**Official recommendations**

Dear reviewers,

The authors' responses come in. Could you take a look at them and then submit your recommendations?

Thanks,
Gang

---

> ### Public Comment · ~Shuo_Chen8 · 2022-07-19
> **Weakly Accept**
>
> Dear AE,
>
> I have read the author's reply. The authors solved my first two concerns, but my third concern about the convergence behavior remains (which is also raised by Reviewer Gw8t). I would like to recommend a "weakly accept".
>
> Thanks!
>
> Best,

---

### Decision · Action_Editors · 2022-07-28

**Recommendation:** Accept with minor revision

**Comment:**

All 3 reviewers recommended weak accept after the rebuttal because the authors have successfully addressed most concerns. I think the paper can be accepted for publication at TMLR after a minor revision.

There are 2 concerns not yet well addressed:

> Another issue is the experimental setting based on semi-synthetic datasets. It is questionable if treating one class as normal and others as abnormal from the CIFAR-10, Fashion MNIST, and Asirra datasets reflects actual anomaly detection applications in the real world.

> The criterion of convergence is not given.

Regarding the first concern, the argument that other papers also did so is rather not a good answer (at least to me). Please be scientific and consider whether other papers are correct or not on this point. What is missing from other papers and this paper is what underlying assumption you made in unsupervised learning without any fitting target. Specifically, do you need any kind of the cluster assumption, the manifold assumption, or in general the low-density separation assumption for unsupervised modeling of the given data? For your proposed method to work, do you need the normal/anomalous data to form clusters or low-density regions? Real-world anomalous data may not possess any cluster structure like "other classes" in multi-class benchmark datasets. This is what our reviewer was worried about and what you should carefully discuss about in the camera-ready paper.

Regarding the second concern shared by all 3 reviewers, I think it's fine to mention the difficulty of such analysis given the currently available tools in math/theory. However, please be aware that when the loss converges, the model being trained may not converge, whenever your optimization problem is non-convex. This may be the case if the model jumps from a solution to another equally good solution at every update. Thus, a converged loss only means your optimization achieves an equilibrium of training but not a converged model itself. Please remove the misleading claims related to your Answer 12 in the camera-ready paper. Thanks!

---

> ### Author Response · Authors · 2022-08-03
> **Addressed final comments in the revised manuscript.**
>
> We thank all reviewers and the action editor for their valuable comments on our work.
>
> **Answer 1**: First, we would like to highlight that we demonstrate the outperformance of SRR not only on semi-synthetic datasets (CIFAR-10, Fashion MNIST, and Asirra) but also on real-world anomaly detection datasets (MVTec, Thyroid, KDD, and Arrhythmia). More specifically for those real-world anomaly detection datasets, there are no cluster, manifold, or low-density separation assumptions. Normal and anomalous labels are directly determined by the real-world data generation processes as we have: (i) manufacturing anomalies (MVTec), (ii) healthcare anomalies (Thyroid, Arrhythmia), and (iii) network anomalies (KDD).
>
> In addition to the experimental results on multiple real-world anomaly detection datasets, we also show the superiority of SRR on semi-synthetic datasets (also can be interpreted as semantic anomaly detection datasets) as they are widely used for the state-of-the-art anomaly detection model comparisons. Note that we show the superiority of SRR on both real-world and semantic anomaly detection datasets.
>
> We have clarified these points in the revised manuscript.
>
>
> **Answer 2**: We acknowledge this comment. In the revised manuscript, we have clarified the convergence parts clearer as follows:
> - We train the SRR frameworks until the representation learner’s loss converges (if no improvement is observed in the representation learner’s loss for 5 epochs).  We use this as the criteria of the convergence (i.e., finalizing the model training as described in Line 9 in Algorithm 1).
> - Note that the convergence of the representation learner’s loss does not guarantee the convergence of the entire SRR frameworks.
> - Empirically, when the representation learner’s loss converges, the data refinement block is also converged as shown in Figure 7 and 11.

---

> > ### Comment · Action_Editors · 2022-08-05
> > **Some clarification**
> >
> > Although I have approved the camera ready submission, I'd like to give some of my thoughts on Answer 1.
> >
> > To be clear, it is different between the two cases, where we cannot verify a dataset follows an assumption, and where we can verify a dataset doesn't follow an assumption. So I am not sure about the claim "for those real-world anomaly detection datasets, there are no cluster, manifold, or low-density separation assumptions".
> >
> > In particular, one-class support vector machine and support vector domain description both need low-density separation. It guarantees there is no data path that connects between a region of normal data and a region of abnormal data in the feature space under consideration. If there is such a path so that such two regions are geometrically connected in the feature space, how can we confidently "define" one end of the path to be normal and the other end of the path to be abnormal?
> >
> > To me, the major contribution of this work is to improve the quality of the learned semantic feature space. To better learn the semantic feature space, some but not all abnormal data are included in one-class classification for diversity, so an ensemble of such one-class classification results is needed for robustness. In the final step of training, the diversity is not important any more, and a single one-class classifier is trained with only identified normal data for performance. I personally think this is because in the final semantic feature space, by employing one-class classification, the proposal implicitly implements low-density separation between normal data (included for training) and abnormal data (excluded for training). This can also explain why such a final step is needed, as excluding abnormal data for training is the correct way to implement low-density separation between normal and abnormal data in one-class classification problems and algorithms. The only difference is in which space low-density separation holds, i.e., the predefined kernel feature space vs. the carefully learned semantic feature space.
> >
> > Last but not least, in discriminative machine learning, low-density separation for classification has the same role as smoothness of the target function for regression or function approximation. We can only avoid low-density separation completely by using generative machine learning such as Gaussian mixture models. Thus, the aforementioned claim "for those real-world anomaly detection datasets, there are no cluster, manifold, or low-density separation assumptions" may not be the case. Just my personal opinion.
> >
> > PS, it seems to be true that the cluster assumption is not needed at all, because I can hardly imagine the anomaly form clusters on real-world anomaly detection datasets!

---

> > > ### Author Response · Authors · 2022-08-05
> > > **Thank you for the thoughtful comments**
> > >
> > > We acknowledge that the latest answer is somewhat confusing. We would like to highlight that our experiments are not only done on the semi-synthetic datasets but also done on real-world anomaly detection datasets. We also acknowledged that we cannot verify whether those real-world datasets follow the cluster, manifold or low-density separation assumptions.
> > >
> > > As you mentioned in the comment, the proposed methods (SRR) try to learn better representations for low-density separation between normal and abnormal samples. But the low-density separation is not explicitly given in the real-world anomaly detection datasets and the superior methodologies can discover that low-density separation more efficiently. In that point of view, we would like to note that the proposed method can be applicable not only to somewhat obvious low-density separation cases (semi-synthetic data), but also to implicit low-density separation cases (real-world anomaly detection data).